# BOOSTR: A Dataset for Accelerator Control Systems

Diana Kafkes *,† and Jason St. John †

Fermi National Accelerator Laboratory, Batavia, IL 60510, USA; stjohn@fnal.gov
* Correspondence: dkafkes@fnal.gov
† These authors contributed equally to this work.

**Abstract:** The Booster Operation Optimization Sequential Time-series for Regression (*BOOSTR*) dataset was created to provide a cycle-by-cycle time series of readings and settings from instruments and controllable devices of the Booster, Fermilab's Rapid-Cycling Synchrotron (RCS) operating at 15 Hz. *BOOSTR* provides a time series from 55 device readings and settings that pertain most directly to the high-precision regulation of the Booster's gradient magnet power supply (GMPS). To our knowledge, this is one of the first well-documented datasets of accelerator device parameters made publicly available. We are releasing it in the hopes that it can be used to demonstrate aspects of artificial intelligence for advanced control systems, such as reinforcement learning and autonomous anomaly detection.

**Keywords:** dataset; artificial intelligence; machine learning; accelerator control systems; anomaly detection

## 1. Introduction

Tuning and controlling particle accelerators is both challenging and time-consuming. Even marginal improvements to accelerator operation can translate very efficiently into improved scientific yield for an experimental particle physics program. The data released here were collected in the hopes of achieving improvement in precision for the Fermilab Booster gradient magnet power supply (GMPS) regulatory system, which is detailed below.

The Fermilab Booster receives a 400 MeV proton beam from the linear accelerator and accelerates it to 8 GeV through synchronously raising accelerator cavity radiofrequency and instigating a controlled magnetic field to steer the beam with combined-function bending and focusing electromagnets, known as gradient magnets. These magnets are powered by the GMPS, which operates on a 15 Hz cycle between a minimum current (at injection) and a maximum current (at beam extraction). The GMPS is realized as four power supplies, evenly distributed around the Booster, and each powers one fourth of the gradient magnets. The role of the GMPS regulator is to calculate and apply small compensating offsets in the GMPS driving signal in order to improve the agreement of the resulting observed minimum and maximum currents with their set points. Without regulation, the fitted minimum of the magnetic field may vary from the set point by as much as a few percent.

At beam injection, a perturbation of only a percent is enough to significantly decrease beam transfer efficiency and thereby reduce the beam flux ultimately available to the high-energy particle physics experiments run at the lab. Disturbances to the magnet current can occur due to ambient temperature changes, other nearby high-power pulsed radio-frequency systems, and electrical ground movement induced by power supplies of other particle accelerators at the complex. The current GMPS regulation involves a

proportional-integral-derivative (PID) control scheme (see Figure 1 for schematic). The regulator calculates estimates for the minimum and maximum currents of the offset-sinusoidal magnetic field from the previous 15 Hz cycle. These values are then used to adjust the power supply program and decrease systemic error in the next cycle's current, such that it more closely matches the set point. Presently, the PID system achieves regulation errors corresponding to roughly 0.1% of the set value.

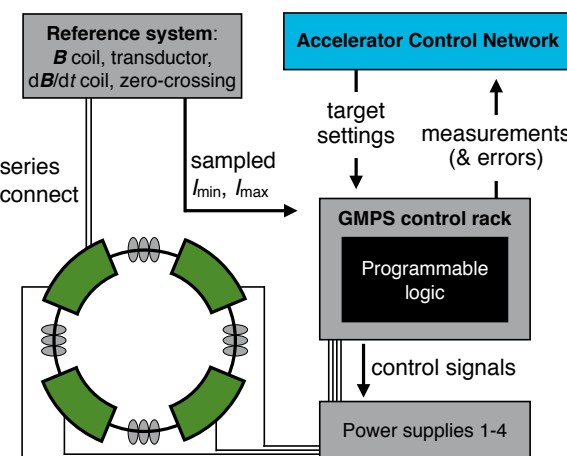

**Figure 1.** Overview of current GMPS control system [1]. Presently, a human operator specifies a target program for `B:VIMIN` and `B:VIMAX`, the GMPS compensated minimum and maximum currents, respectively, via the Fermilab Accelerator Control Network, which is transmitted to the GMPS control board.

Although some 200,000 entries populate the device database of Fermilab's accelerator control system [2], the 55 device value time series presented here in *BOOSTR* [3] were collected in accordance with suggestion by Fermilab accelerator subject matter experts (SMEs). These values include PID controller settings and readings as well as values that exhibit correlations with GMPS power supply perturbations. The full data were collected during two separate periods: from 3 June 2019 to 11 July 2019—when the accelerator was shut down for regular maintenance—and from 3 December 2019 to 13 April 2020—when the accelerator was shut down in response to the COVID-19 pandemic.

Data from a single day of *BOOSTR* were previously described in a Datasheet [4]. A proof-of-concept paper [1] (submitted to *Physical Review Accelerators and Beams*) used this subset of *BOOSTR* to develop a workflow for training a reinforcement learning agent to regulate GMPS via a field-programmable gate array (FPGA). Relative to the original datasheet [4], this manuscript is expanded with more SME input, describes more than 100 times more data, and includes documentation of validation not presented in the original datasheet.

## 2. Methods

### 2.1. Collection Process

A data collection node was created and set to request data at 15 Hz from the Data Pool Manager of the Accelerator Control Network (ACNET) [2]. The created scheme involved front-end nodes, each managing their respective devices, replying with timestamped values at the stated rate barring differences of clock speed, input–output (I/O) lag time variations due to network traffic fluctuations, and higher-priority interruptions from competing processes on the front-end node. These inconsistencies were later addressed through a time-alignment process discussed in the Data Processing Section. The collection node stored the data in a circular buffer approximately 10 days deep.

A Python script managed by a nightly cron job polled the data collection node for the most recent midnight-to-midnight 24 h of timestamped data for each of the 55 time series identified by the SMEs. A second cron-managed script did the same for relevant accelerator

control events issued in the same period. These event data correspond to important cycles achieved through controlling the devices at the accelerator. Event data were requested by a separate data collection node.

Each day's data harvest was originally stored in HDF5 (Hierarchical Data Format Version 5) files. Any data instances missing from the parquet files released here were not included in the original data buffers from which this dataset was drawn.

*2.2. Data Processing*

Each instance was created through a concatenation of each device's timestamp data table within every HDF5 file and then saved in parquet format. A similar procedure was undertaken for one of the accelerator control event signals polled, `Event0C`, as its broadcast is synchronized with the GMPS magnetic field minimum. `Event0C` was collected to correct a problem in the observed sampling frequency: there was an issue of the sampling of each device being nominally at 15 Hz, but in reality synchrony was demonstrably imperfect, and the time intervals between successive timestamps display varying lag.

Since `Event0c` serves as the baseline or heartbeat of the Booster at approximately 15 Hz and is synchronized with the smoothly varying electrical current GMPS regulates, we used `Event0c` to time-align our data. The alignment approximates the data available to the GMPS regulator operating in real time. We used the GMPS-synchronized `Event0C`'s timestamp as the moment to begin forward inference, taking the value for each device time series that had the most recent corresponding timestamp. In practice, this required timestamp-sorted series for each device and finding the most recent device value, relative to `Event0c` timestamp, in a lookback window equal to the maximum interval between device timestamps (necessarily excluding the five month gap between our two data collection periods). This time-alignment step was run over the whole dataset in multiple parallel processes using Apache Spark.

Notably, the data recorded for `Event0c` were missing the period from 1 to 11 July 2019. Therefore, aligning on this variable discarded some of the data collected during our first period of collection.

## 3. Data Description

The data release is stored on Zenodo [3]. Each instance is a zip compressed parquet of one of the 55 aligned time series with columns corresponding to the aligned time stamp, original time stamp, difference between time stamps, and the reading/setting value [3]. The original timestamp and time difference is included to demonstrate the mechanics of our alignment process and enable a check for reproducibility. All timestamps are in Greenwich Mean Time (UTC).

Our data release contains device data from each of the four gradient magnet power supplies, the GMPS PID regulator, and the main injector, where the beam is directed after acceleration via the Booster. Minimum and maximum current information readings and settings, the feedback and transductor gain settings, and the feed-forward start trigger are collected as part of the current PID regulation scheme. The "inhibit" value controls whether the GMPS regulator accepts setting changes for parameters other than the minimum and maximum current, such as the gain settings (any positive value will prevent changes). Additionally, $\dot{\vec{B}}$, the rate of change of the magnetic field, is recorded as a proxy for the magnetic field we are interested in regulating. Timing information derived from $\dot{\vec{B}} = 0$ synchronizes the current PID regulator system.

We acknowledge that the ACNET parameter names are by no means standardized across different particle accelerators and that they will appear especially abstruse for those well-versed in control systems who are new to working with accelerators. In Table 1, we detail explanations of each of the parameters read from devices (devices whose first letter is followed by:) and indicate whether the device setting was included in the dataset (devices whose first letter is followed by _ and appear in the Setting column), since describing these corresponding pairs would be redundant. In Figures 2 and 3, we visualize metadata trends

for each "nonconstant" parameter in each data collection period (see Table 2 for a list of values we considered to be virtually unchanging within the two periods) and also provide the mean and standard deviation of device readings across the two collection periods in Table 1. Furthermore, Table 1 includes dates missing in the data recorded for each reading. As a reminder, the data were collected during two separate periods: from 3 June 2019 to 30 June 2019 (July is missing due to time-alignment with Event0C) and from 3 December 2019 to 13 April 2020. Finally, in Figure 4 we demonstrate the centrality of each recorded parameter with a heatmap of histogram values.

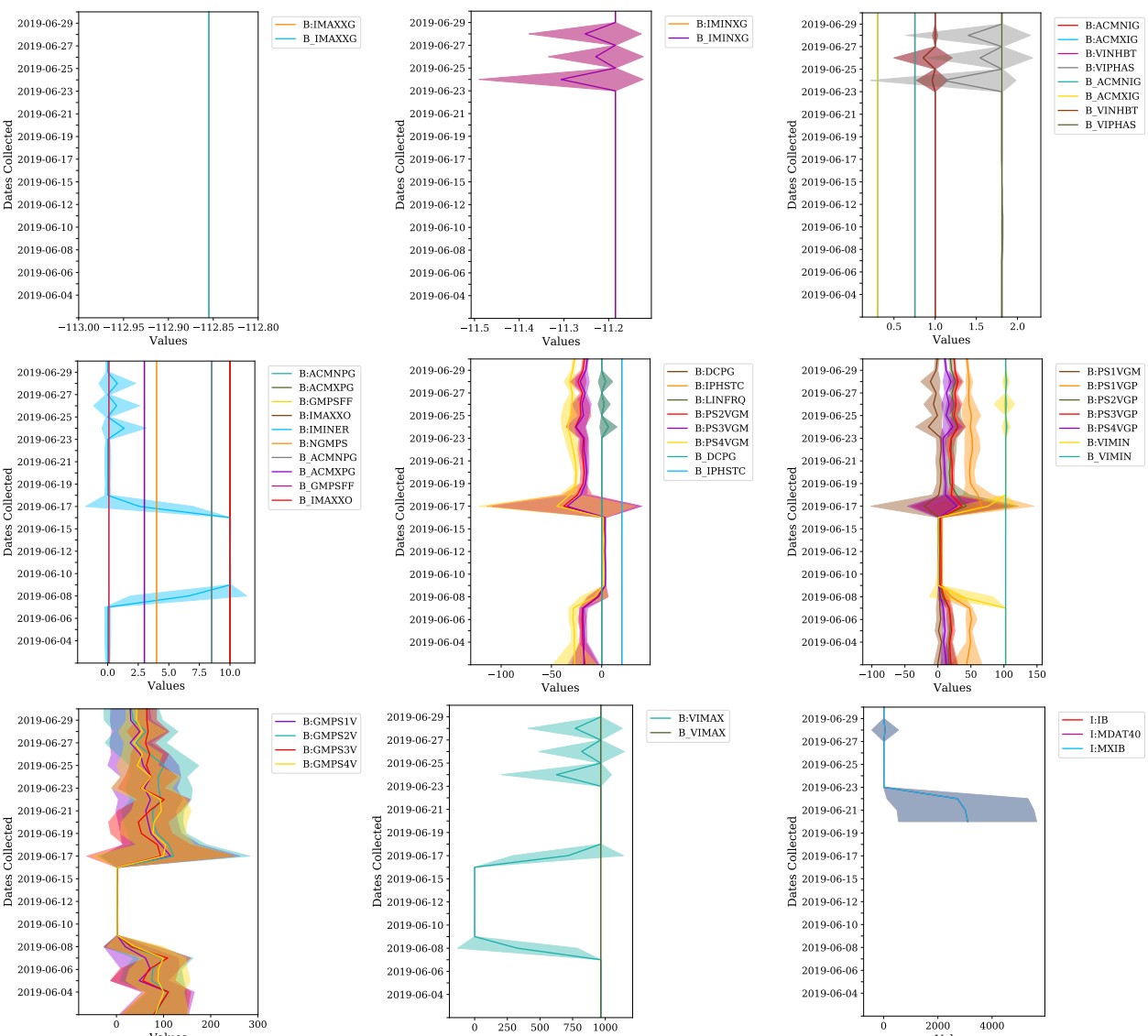

**Figure 2.** Metadata variable trends for Period 1: 3 June 2019 to 30 June 2019. Event0c was missing the period from 1 to 11 July 2019, so alignment on this variable discarded any data collected from this time period. The graphs show the mean for each variable on the given date and shades in the standard deviation of that variable on that date.

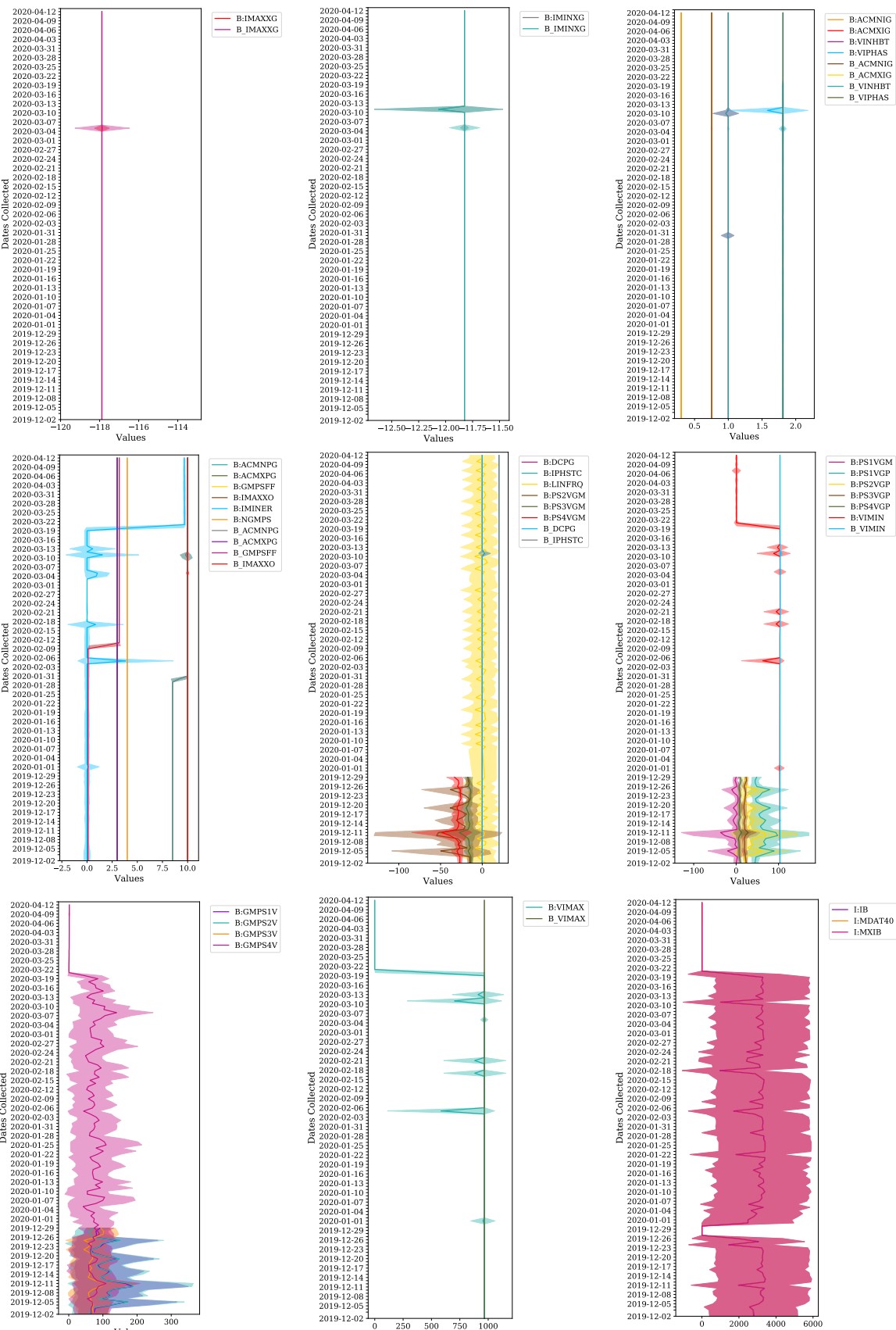

**Figure 3.** Metadata variable trends for Period 2: 2 December 2019 to 13 April 2020. Any data instances missing from the parquet files released here were not included in the original data buffers from which this dataset was drawn. The graphs show the mean for each variable on the given date and shades in the standard deviation of that variable on that date.

**Table 1.** Description of the BOOSTR dataset parameters. Here "GMPS" denotes the gradient magnet power supplies (1–4), "MI" means main injector, "MDAT" refers to Fermilab's machine data communications protocol. Device parameters that begin with B relate to the accelerator Booster, whereas device parameters that begin with I relate to the main injector. Parameter mean and standard deviation have been truncated to two decimal points.

| Parameter | Details (Units) | Setting | Mean (Std) | Missing Dates |
|---|---|---|---|---|
| B:ACMNIG | Min AC integral feedback gain | B_ACMNIG | 0.75 (0) | 2019: 6/14, 12/12 |
| B:ACMNPG | Min AC proportional feedback gain | B_ACMNPG | 9.1968 (0.75) | 2019: 6/14, 12/12 |
| B:ACMXIG | Max AC integral feedback gain | B_ACMXIG | 0.30 (0) | 2019: 6/14, 12/12 |
| B:ACMXPG | Max AC proportional feedback gain | B_ACMXPG | 3.00 (0) | 2019: 6/14, 12/12 |
| B:DCIG | DC integral feedback gain | B_DCIG | 0 (0) | 2019: 6/14, 12/12 |
| B:DCPG | DC proportional feedback gain | B_DCPG | 0.10 (1.35) | 2019: 6/14, 12/12 |
| B:GMPS1V | GMPS1 output voltage (V) | | 81.82 (89.56) | 2019: 6/14, 12/12, 12/30–31 and 2020: 1/01–4/12 |
| B:GMPS2V | GMPS2 output voltage (V) | | 85.29 (96.00) | 2019: 6/14, 12/12, 12/30–31 and 2020: 1/01–4/12 |
| B:GMPS3V | GMPS3 output voltage (V) | | 63.67 (61.19) | 2019: 6/14, 12/12, 12/30–31 and 2020: 1/01–4/12 |
| B:GMPS4V | GMPS4 output voltage (V) | | 61.08 (65.19) | 2019: 6/14, 12/12 |
| B:GMPSBT | $\partial B/\partial t$ offset trigger (s) | B_GMPSBT | 0 (0) | 2019: 6/14, 12/12 |
| B\|GMPSSC | Binary status control of GMPS | | N/A (N/A) | 2019: 6/07, 6/12, 6/14–15 and 2020: 1/18, 3/08, 3/15 |
| B:GMPSFF | Feedforward start trigger (s) | B_GMPSFF | 1.32 (1.52) | 2019: 6/14, 12/12 |
| B:IMAXXG | Max transductor gain (A/V) | B_IMAXXG | −117.12 (1.80) | 2019: 6/14, 12/12 |
| B:IMAXXO | Max transductor offset (A) | B_IMAXXO | 10.00 (0) | 2019: 6/14, 12/12 |
| B:IMINXG | Min transductor gain (A/V) | B_IMINXG | −11.73 (0.23) | 2019: 6/14, 12/12 |
| B:IMINXO | Min transductor offset (A) | B_IMINXO | 0 (0) | 2019: 6/14, 12/12 |
| B:IMINER | Discrepancy from setting at min (0.1 A) | | 1.93 (3.86) | 2019: 6/14, 12/12 |
| B:IMINST | $\partial B/\partial t$ sample off | B_IMINST | 0 (0) | 2019: 6/14, 12/12 |
| B:IPHSTC | Phase regulator time constant | B_IPHSTC | 20.00 (0.01) | 2019: 6/14, 12/12 |
| B:LINFRQ | 60 Hz line frequency offset (mHz) | | −0.44 (16.31) | 2019: 6/03–7/11, 12/12 12/30–31 and 2020: 1/01–06 |
| B:NGMPS | Number of GMPS suppliers | | 4.00 (0) | 2019: 6/14, 12/12 |
| B:PS1VGM | GMPS1 V− to ground (V) | | −2.30 (23.56) | 2019: 6/14, 12/12, 12/30–31 and 2020: 1/01–4/12 |
| B:PS2VGM | GMPS2 V− to ground (V) | | −21.29 (27.52) | 2019: 6/14, 12/12, 12/30–31 and 2020: 1/01–4/12 |
| B:PS3VGM | GMPS3 V− to ground (V) | | −15.13 (14.11) | 2019: 6/14, 12/12, 12/30–31 and 2020: 1/01–4/12 |
| B:PS4VGM | GMPS4 V− to ground (V) | | −26.27 (17.22) | 2019: 6/14, 12/12, 12/30–31 and 2020: 1/01–4/12 |
| B:PS1VGP | GMPS1 V+ to ground (V) | | 52.00 (34.82) | 2019: 6/14, 12/12, 12/30–31 and 2020: 1/01–4/12 |
| B:PS2VGP | GMPS2 V+ to ground (V) | | 26.53 (30.53) | 2019: 6/14, 12/12, 12/30–31 and 2020: 1/01–4/12 |
| B:PS3VGP | GMPS3 V+ to ground (V) | | 20.17 (13.74) | 2019: 6/14, 12/12, 12/30–31 and 2020: 1/01–4/12 |
| B:PS4VGP | GMPS4 V+ to ground (V) | | 9.14 (12.75) | 2019: 6/14, 12/12, 12/30–31 and 2020: 1/01–4/12 |
| B:VIMAX | Compensated max GMPS current (A) | B_VIMAX | 772.43 (385.98) | 2019: 6/14, 12/12 |
| B:VIMIN | Compensated min GMPS current (A) | B_VIMIN | 83.20 (40.68) | 2019: 6/14, |
| B:VINHBT | Inhibit value | B_VINHBT | 1.00 (0.04) | 2019: 6/14, 12/12 |
| B:VIPHAS | GMPS ramp phase wrt line voltage (rad) | B_VIPHAS | 1.80 (0.14) | 2019: 6/14, 12/12 |
| I:IB | MI lower bend current (A) | | 2275.71 (2571.03) | 2019: 6/03–19, 12/12 |
| I:MDAT40 | MDAT measured MI current (A) | | 2400.57 (2576.37) | 2019: 6/03–7/11, 12/12 |
| I:MXIB | Main dipole bend current (A) | | 2279.68 (2564.53) | 2019: 6/03–11, 6/13–19, 12/12 |

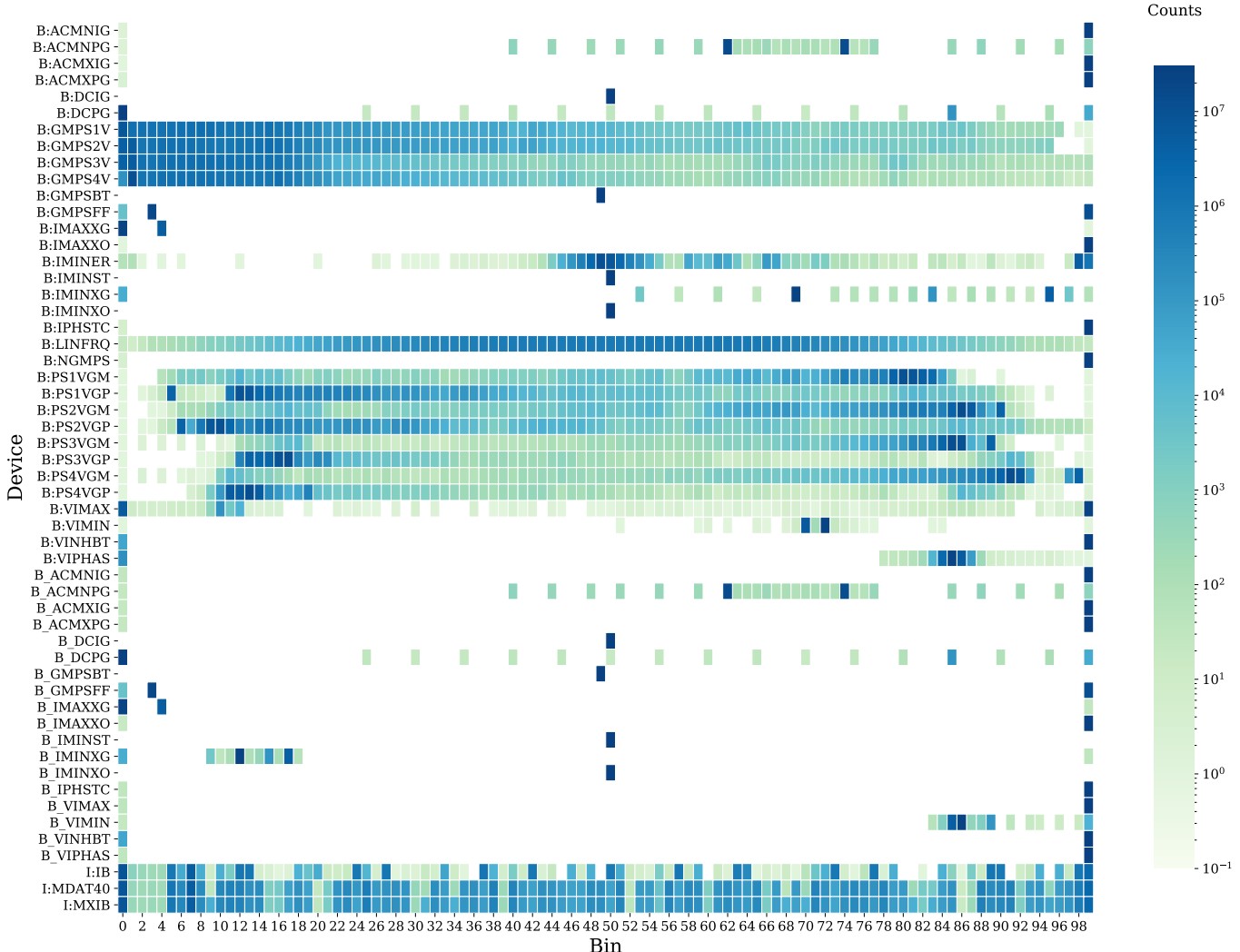

**Figure 4.** Heatmap of histogram distributions for each reading and setting variable. Histograms contain one hundred bins sampled on even intervals between the minimum and maximum value of each variable. This is only meant to characterize the centrality of each recorded value. See Figures 2 and 3 for actual metadata value ranges.

**Table 2.** Summary of nearly constant variables in both periods of data collection. Here "nearly constant" denotes variables having a standard deviation less than $10^{-5}$ across both periods.

| Device | Setting | Constant Value |
|---|---|---|
| B:DCIG | B_DCIG | 0 |
| B:GMPSBT | B_GMPSBT | 0.0003 |
| B:IMINST | B_IMINST | 0 |
| B:IMINXO | B:IMINXO | 0 |

Additionally, we provide the PID regulator status values B|GMPSSC (ACNET status parameters include |), whose 16 bits contain various motherboard states. Here we are concerned with bit 3, which indicates whether or not the GMPS regulator was on (1), and bit 7, which indicates whether the Booster is in its normal alternating current (AC) mode (1) or "coasting beam" direct current (DC) mode at constant beam energy (0). Unlike the rest of the devices, this status value is presently recorded at only 1 Hz because it was not included in our initial data node request and was relegated to an archived data node at a lesser frequency. While the same time-aligning described above was applied to align B|GMPSSC, due to the slow sampling rate, we caution the user to refer closely to the original timestamp

such that they might make decisions about whether to use data when GMPS was off and to inform them of potential problems when interpolating in a region immediately before or after a status change. See Figure 5 for more details on B|GMPSSC values.

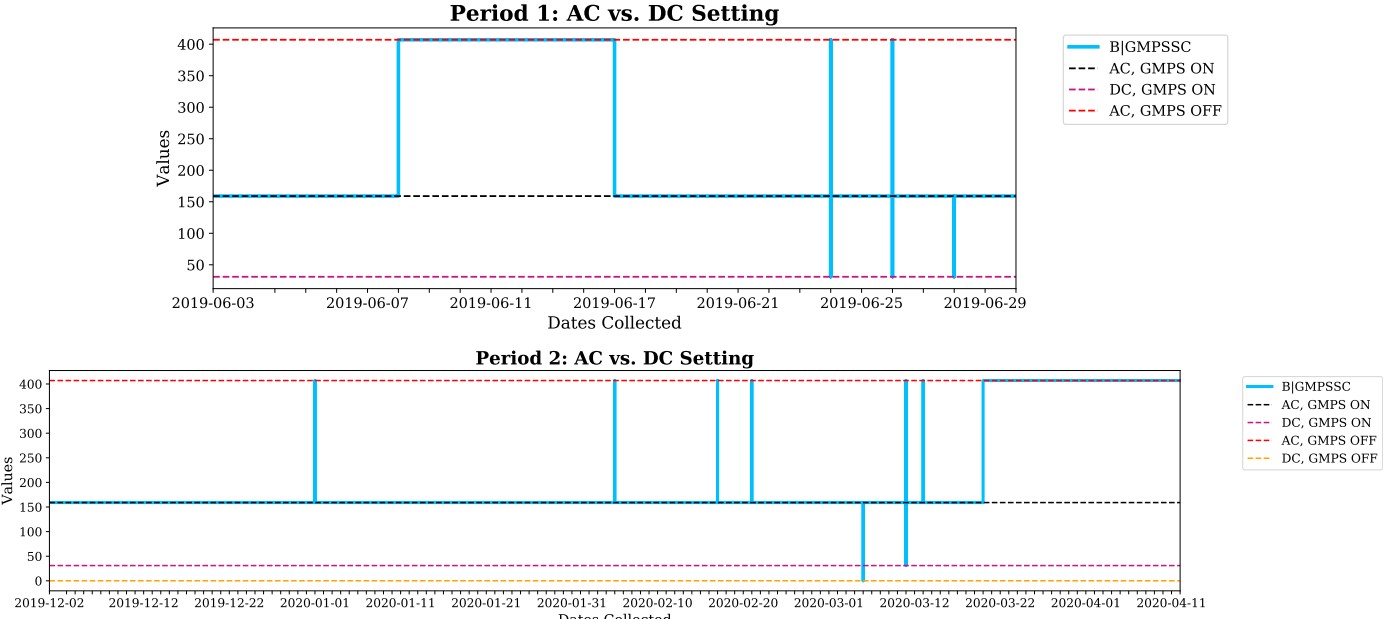

**Figure 5.** Daily values of B|GMPSSC (should be interpreted as taking the mode for each day) whose bits encode relevant Booster statuses.

## 4. Technical Validation

In order to verify the quality of this dataset, we pored over the electronic logbook (Elog) [5] that Fermilab Booster technicians and operators use to record changes to device settings as well as observations while in operation. We used these Elog entries to authenticate our data's viability across timescales. First, we used the Elog to corroborate expert acknowledgment of the major spikes observed in Figures 2 and 3. These outlier changes, typically seen in the value's mean and standard deviation, represented major changes made on that specific day, including when the Booster was switched from alternating current (AC) to direct current (DC) mode (see Figure 5) as well as when the GMPS regulator was turned off altogether. These reconciliations are presented in Table 3.

Furthermore, we pinpoint changes in the AC vs. DC settings according to the Elog [5] for 24 June 2019 and 11 March 2020 in Figure 6. Here, applying a bitmask reveals that a B|GMPSSC value of 159 indicates AC mode/GMPS on, while 31 indicates DC mode/GMPS on, and 407 indicates AC mode/GMPS off. In this figure, the plotted timestamps were offset to Central Time (UTC-5) in order to align with times given in the Elog, which were not recorded in UTC. On 24 June, the trace of B|GMPSSC clearly shows GMPS regulation briefly switching off before commencing DC studies from 8:00 AM–6:00 PM with a value of 31, then being turned back to 159. On 11 March, B|GMPSSC is at 159 before 6:00 AM, off at 407 from 6:00–9:50 AM, and then is set to AC mode from 9:50 AM–12:45 PM, to DC mode from 12:45–3:49 PM, and back to AC mode for the rest of the day, as per the Elog [5]. The close correspondence of these changes in our data to the recorded actions and observations of Booster personnel boost our confidence in the quality and relevance of the collected dataset.

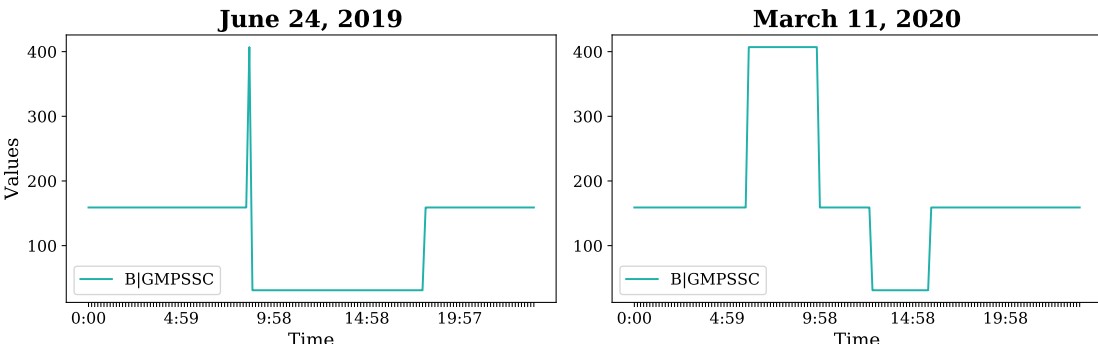

**Figure 6.** Values of status `B|GMPSSC` corresponding to Table 3 entries for 24 June 2019 and 11 March 2020 (timestamps were put in Central Time to align with Elog). Recall: a value of 159 indicates AC study/GMPS on, 31 indicates DC study/GMPS on, and 407 indicates AC study/GMPS off. These traces display this value at a much greater granularity than Figure 5.

Additionally, we plot settings changes on March 10 and 11 documented in the Elog Figure 7. The blip in `B:ACMNPG` from 6.5 to 13.5 is visible as is the slight decrease in `B_VMIN` around 4:00 PM CST, which were mentioned in Table 3.

**Table 3.** Summary of Booster-related electronic log (Elog) [5] entries corresponding to spikes in Figures 2 and 3. Original Central Time times are given with values in parenthesis designating UTC. Here, "RF" denotes radiofrequency.

| Date | Elog Entry Summary |
|---|---|
| 6/8/2019 | GMPS in AC mode until 8:00 AM (13:00) on 6/08, then switched off until 6/17 |
| 6/10/2019 | GMPS was locked/tagged out for outage, West Booster gallery RF off from 8:30–10:00 (13:30–3:00) for work |
| 6/17/2019 | GMPS turned back on and put in AC mode |
| 6/22/2019 | High energy physics beam turned off at 8:00 PM (1:00 <sup>+1</sup>) (GMPS remained in AC mode) |
| 6/24/2019 | DC studies from 8:00 AM–6:00 PM (13:00–23:00), back to AC mode |
| 6/26/2019 | Alternated between AC and DC mode, GMPS off for 30 min around 5:30 PM (22:30) |
| 6/27/2019 | GMPS in AC mode all day, but removing certain study events caused bias to creep up, eventually tripping the RF |
| 6/28/2019 | Alternated between AC and DC mode |
| 12/8/2019 | `B_VMIN` adjusted, GMPS in AC mode |
| 12/12/2019 | AC mode, operators reset virtual machine environment locally |
| 12/28–12/30/2019 | No beam from injector, GMPS in AC mode |
| 12/31/2019 | Booster injection back, GMPS in AC mode |
| 1/1/2020 | GMPS off for 15 min around 9:30 AM (14:30), in AC mode for rest of day |
| 2/4/2020 | RF sparking in gallery due to reverting of RF capture settings, GMPS in AC mode |
| 2/5/2020 | GMPS off from 6:00 AM–3:30 PM (11:00–20:30), then in AC mode for rest of day |
| 2/6/2020 | Lowered beam intensity to users, but GMPS was in AC mode all day |
| 3/5/2020 | Beam tails were large, so turned `B_VMIN` down |
| 3/10/2020 | GMPS in AC mode all day, `B:ACMNPG` changed from 6.5 to 13.5, `B_VMIN` decreased from 103.440 to 103.420 |
| 3/11/2020 | GMPS in AC mode from 12:00 AM–6:00 AM (5:00–11:00), off from 6:00–10:00 AM (11:00–15:00), then alternated between AC and DC mode, `B_VMIN` adjusted from 103.418 to 103.386 |
| 3/13/2020 | GMPS off from 9:30 AM–11:00 AM (14:30–16:00), back on and put in AC mode |
| 3/20/2020 | Booster turned off on account of Covid-19 pandemic at 12:00 PM (17:00) |

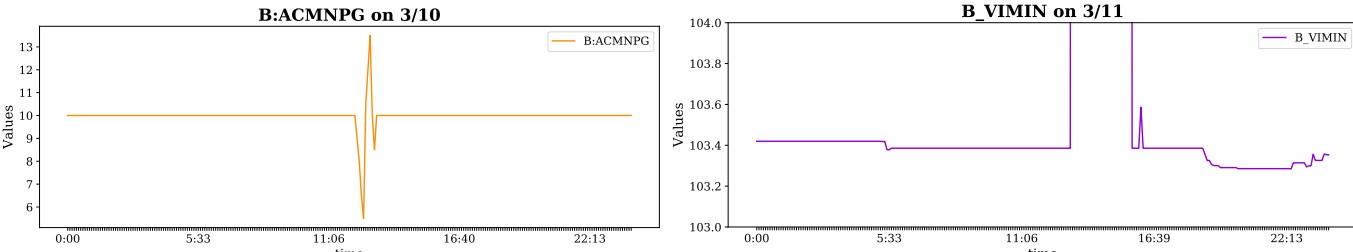

**Figure 7.** Switches corresponding to Table 3 entries for 10 March 2020 and 11 March 2020: `B:ACMNPG` changed from 6.5 to 13.5 and `B_VIMIN` decreased from 103.418 to 103.386 (timestamps were put in Central Time to align with Elog). The sudden large increase in `B_VIMIN` from 12:45–3:49 PM CST to a value near 120 corresponds to the DC mode observed in Figure 6.

## 5. Usage Notes

*BOOSTR* can be used to develop a wide variety of machine learning based approaches for accelerator network tuning and control. Possibilities include training various control networks for accelerator regulation, constructing "digital twins" of the Fermilab Booster regulator's control environment, designing Bayesian optimization, and developing anomaly detection/categorization capabilities. Please note: there are no legal or ethical ramifications of using these data as they were collected from a machine, and not collected from or representative of people.

In the future, the dataset could feasibly be expanded to include more of the 200,000 available ACNET system parameters and therefore be used to control, mimic, or monitor further aspects of the particle accelerator complex. Since many common challenges exist across different accelerator facilities, open datasets such as this one are crucial for the further development of cross-facility machine learning methods.

The preprocessing code and a simple sample notebook for data loading are available online: https://github.com/dkafkes/BOOSTR. When using *BOOSTR* data, the authors recommend ordering by time immediately, as the parquet files do not store the data entries sequentially [3]. See example data loading notebook for more detail on how to import the data for use.

**Author Contributions:** J.S.J. created the data collection script, and set up and maintained the cron jobs to record the data in HDF5 files. D.K. migrated the data from on-premise storage to the cloud, wrote the preprocessing and time-alignment code, and validated the data. Both authors reviewed this manuscript. Both authors have read and agreed to the published version of the manuscript.

**Funding:** This dataset was created as part of the "Accelerator Control with Artificial Intelligence" Project conducted under the auspices of the Fermilab Laboratory Directed Research and Development Program (Project ID *FNAL-LDRD-2019-027*). The manuscript has been authored by Fermi Research Alliance, LLC under Contract No. DE-AC02-07CH11359 with the U.S. Department of Energy, Office of Science, Office of High Energy Physics and is registered at Fermilab as Technical Report Number *FERMILAB-PUB-21-005-SCD*.

**Institutional Review Board Statement:** Not applicable.

**Informed Consent Statement:** Not applicable.

**Data Availability Statement:** The data presented in this study are openly available https://doi.org/10.5281/zenodo.4382663.

**Acknowledgments:** We are extremely grateful for Brian Schupbach and Kent Triplett for lending their Booster technical expertise, without which we could not have validated our dataset. Additionally, we would like to acknowledge Burt Holzman for guidance on getting set up in the cloud, and the help of the Databricks federal support staff, in particular Sheila Stewart. Furthermore, we would like to recognize Aleksandra Ciprijanovic and Jovan Mitrevski for useful discussions and a careful reading of this manuscript.

**Conflicts of Interest:** The authors declare no competing interests.

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
