# Peer review of "BOOSTR: A Dataset for Accelerator Control Systems"

_data, 2021_

Round 1

Reviewer 1 Report

see attached file

Reviewer 2 Report

The authors present a dataset, BOOSTR, of time series data from 55 devices related to the high-precision regulation of the Booster’s Gradient Magnet Power Supply (GMPS) at Fermilab’s Rapid-Cycling Synchrotron. The data set is proposed for the aid of designing artificial intelligence approaches for advanced accelerator control systems.

The data is presented as two continuous segments, June 3, 2019 to June 30, 2019 and December 3, 2019 to April 13, 2020. These are impressively long periods of time and it will be interesting to design controllers using two data sets with 6 months of separation to see how and if the system significantly changed with time, which is an important issue for most accelerators.

Regarding the PID controller, does the data only provide a status of whether the PID is on or off, or does it actually provide the output values of the PID controller? For any kind of system identification/characterization the actual PID control time series data would be very useful and important. If that data is not present in this dataset, if possible, I would recommend that the authors add it at a future date.

Under Section 5, Usage Notes, the authors recommend ordering by time when using the BOOSTR data. I checked the link to data set [3], I would highly recommend that the authors provide a simple python code for download together with their data sets, which a user can start with. This python code should import the data sets, order them by data, and split them up into numpy arrays that are easy to work with. For example, for a system such as

dx/dt = f(x,u,t),    (1)

where x is some physical accelerator parameter that we would like to control and u is a collection of controls/tunable accelerator parameters, a simple python script should be provided that just pulls in x, u, and t as numpy arrays and then users who want to develop ML or RL algorithms for developing controllers for this data for systems of the form (1) can easily start to work with the data instead of wasting a huge amount of time trying to figure out what all of these specific signals are, which the authors are very familiar with, but nobody else in the world is.

I would actually recommend that they authors make several such python codes, that give examples such as: This script pulls in the (x,y) BPM positions of the beam over the entire time period and also the corresponding current settings of magnets M1 and M2, which influence (x,y). That way the data set becomes easily useable for someone who is not expertly familiar with the GMPS and they would be more likely to dig in deeper into the data set.

Round 2

Reviewer 1 Report

NA